# Investigating Plant–Bird Co-Occurrence Patterns in Mediterranean Wetlands: Can They Reveal Signals of Ecosystem Connectivity?

**Mauro Fois** [1] , **Alba Cuena-Lombraña** [1,]*, **Carla Zucca** [2], **Sergio Nissardi** [2] **and Gianluigi Bacchetta** [1]

[1] Centro Conservazione Biodiversità, Dipartimento di Scienze della Vita e dell'Ambiente, Università degli Studi di Cagliari, Viale S. Ignazio da Laconi 11, 09123 Cagliari, Italy; mfois@unica.it (M.F.); bacchet@unica.it (G.B.)

[2] Anthus s.n.c. Studi e Consulenze Ambientali, Via Luigi Canepa 22, 09129 Cagliari, Italy; corymbosa@hotmail.com (C.Z.); nissardi@hotmail.com (S.N.)

\* Correspondence: albacuena@gmail.com or alba.cuena@unica.it

**Abstract:** Interspecific biotic interaction is believed to be a fundamental phenomenon in ecology. However, despite the increasing efforts, interaction mechanisms are still not clearly understood. We compiled a database of 323 birds and 844 vascular plants in 30 wetlands from Sardinia. This was complemented with seed dispersal features and plant structures (suitability for nesting), and with site-level traits, such as wetland surface area, distance from the sea, percentage of open water, protection level, and number of human impacts. The percentage of non-random co-occurrences was then measured, and the relative importance of each trait in determining it was modelled. We found that non-random co-occurrences among sites decreased with the site extent and increase with the percentage of open water, bird zoochory was positively correlated with co-occurrences, nesting birds showed higher rates of co-occurrence than non-nesting birds, and plants with habits suitable for nesting displayed more co-occurrences than the rest of the plants. These results are a small contribution to the complex topic of species co-occurrence and connectivity within an ecosystem. Species co-occurrence is a promising but debatable approach that may provide insightful clues to species interactions within ecological systems.

**Keywords:** community ecology; zoochory; Mediterranean island wetlands; plant–bird interactions; species traits; vegetation structure; wetland conservation

## 1. Introduction

Interspecific biotic interaction is recognised as one of fundamental phenomena in ecology and conservation, because their reduction or loss often affect community composition, structure and functionality and, more generally, the resilience of a habitat [1,2]. Collaborative behaviours in nature are increasingly demonstrated against the competitive theory to play a critical, but overlooked role [3,4]. Among species groups, plant–animal interactions are widespread and essential for ecosystems functioning and maintenance. In general, ecological connectivity through plant–animal interaction should be stronger for those subgroups of birds and plants that are more specialised on each other than for those with weaker network interaction. If the type of interaction is in some cases proven, such as plant–insect pollination mutualisms or herbivory, in other cases, interactions (negative or positive) among species are the result of several factors which are difficult to control or predict. This is the case of plant–bird interactions. Among specific examples, bird and woody plant species richness are often linked to vegetation structural complexity and bird-nesting opportunities [5]. Other interactions, such as frugivory or propagules dispersal, are common positive and neutral interactions that influence the distribution patterns of both groups of species. These interactions are of particular interest in the case of migratory birds—several are waterbirds—which enable plants to disperse for long distances in a

relatively short time [6–8]. Recent discoveries in this sense, such as the extreme ability to disperse non-fleshy-fruited plants [9], has confirmed the need to fill large gaps in this topic. Moreover, plant–bird interactions comprise a wide range of overlooked relationships, such as ingestion and dispersal or the control of arthropod herbivory by birds [10].

Understanding factors influencing patterns of species interaction is a recent major aim of community ecology [11,12]. Interactions might be influenced by the same drivers of spatial patterns in species and habitats diversity, such as natural gradients, elevation, distance from the coast or human-induced environmental changes [13–15]. For instance, historical human presence on several Mediterranean islands acted as a relevant process in disrupting the energy and resource flow and lowering the bird trophic level to one dominated by herbivory and omnivory [16]. In line with this theory but with a different perspective, some studies have stressed that ecological interactions are often lost at a higher rate than species, affecting species functionality and ecosystems services at a faster rate than local species extinctions [17]. For the same reason, habitat fragmentation has been shown to have strong effects on ecological connectivity [13].

Despite the importance of interactions for self-sustaining ecosystems and successful management, only a relatively small proportion of conservation-oriented studies have explicitly considered interactions. The main reason is the high initial cost of conservation studies aimed at unravelling such complex biological networks. In the era of big data, null model analysis has become a standard tool to search for patterns that may reflect processes of community interactions [18]. Null models can be based on the analysis of a presence–absence matrix that allows for a classification of all the unique species' pairs in an assemblage as random, aggregated, or segregated [18–20]. Although inferring ecological interactions from presence–absence data holds a great appeal, the existence of signals of cause-and-effect are still debated. While some recent studies have confirmed such a regular signal (e.g., [21,22]), others have shown that the signal is blurred and diluted in complex networks [23] or even absent [24]. In order to maximize the chance of distinguishing between the influence of environmental preferences and biotic interactions, classic null models were compared or implemented by the introduction of frameworks with species distribution models (SDMs; [11]) and joint species distribution models (JSDM; [25]). Research studies that have attempted to examine species co-occurrences through SDMs and JSDMs were often limited to few species/interactions or to simulated species. Reasons were once again related to the complex model interpretation, the high time required, and the missing data for validation [26]. In synthesis, despite the significant improvements in modelling efforts, compared to early presence–absence null models, the ability to reveal the interaction mechanisms is still controversial and unresolved [23,27,28].

We used a classic null model framework (sensu [1,11]), including information of 323 bird and 844 vascular plant species from a set of 30 wetlands from Sardinia (second largest island of the Mediterranean Basin), to carry out a relatively easily interpretable methodological framework, which includes the following steps: (1) finding significant co-occurrences between plants and birds from a relatively large and feasible set of commonly available presence–absence data; (2) define characteristics related to sites (environmental conditions and level of human disturbance), plants (structural traits, exoticity and dispersal syndrome) and birds (nesting and dispersing); (3) model the site- and species-based number of co-occurrences in response to their respective traits.

Acknowledging that co-occurrence is not evidence of biological interactions, we tested if the signal of such ecological connectivity can be captured in observational data by examining the following hypotheses: Hypothesis 1 (H1), environmental filtering hypothesis: co-occurrences vary among sites according to their environmental diversity and decrease with the presence of threats/impacts; Hypothesis 2 (H2), dispersal hypothesis: bird dispersers show a high number of non-random co-occurrences with plants dispersed by bird zoochory; Hypothesis 3 (H3), vegetation structure hypothesis: birds using plants for nesting show a high number of non-random co-occurrences with plants having habits suitable for nesting.

## 2. Materials and Methods

### 2.1. Study Area and Sampling

The island of Sardinia (Italy) offers the opportunity to cover studies in a diverse set of environments in a reasonable small extent. Specifically, a recent inventory of the Sardinian wetlands found a total number of 2501 wetlands, highly variable in size and typology (www.italiaiswet.it, accessed on 1 March 2022). Most of these wetlands, especially the 706 natural wetlands, covering a total surface area of 365.6 km$^2$, support a rich biodiversity. A recent inventory of strictly hydro- and hygrophilous vascular plants in Italy, identified for Sardinia, 119 species equal to 42.6% and 37.9% of the richness estimated at the Italian and European scale, respectively [29]. Unfortunately, part of them are threatened and unprotected [30,31]. Sardinian wetlands also represent a network of important stopover sites for migratory birds along the Mediterranean flyways [32]. Therefore, the conservation and management of its wetlands is pivotal for many avian species, in particular passeriforms, ducks in winter, and waders in summer and autumn [33].

In order to limit efforts and cover a diverse set of wetland conditions with homogenous data availability, the selection of the study sites considered: (a) wetlands where the available literature was exhaustive to support our field data and/or efforts to validate or complete it were feasible; (b) wetlands differing in, at least, distance from the coast and protection level, covering as large as possible range of typologies. Finally, 30 wetlands were identified. These comprised the eight Ramsar sites, but also sites within national and regional parks, Natura 2000 network or none of them. The distance from the coast ranged from 0 to 33 km (Figure 1).

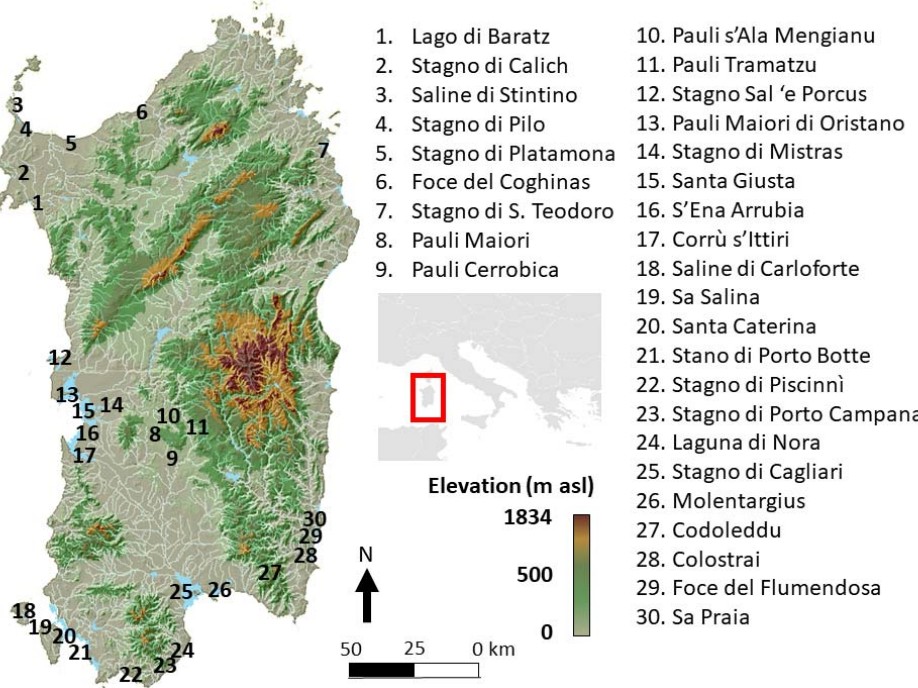

1. Lago di Baratz
2. Stagno di Calich
3. Saline di Stintino
4. Stagno di Pilo
5. Stagno di Platamona
6. Foce del Coghinas
7. Stagno di S. Teodoro
8. Pauli Maiori
9. Pauli Cerrobica
10. Pauli s'Ala Mengianu
11. Pauli Tramatzu
12. Stagno Sal 'e Porcus
13. Pauli Maiori di Oristano
14. Stagno di Mistras
15. Santa Giusta
16. S'Ena Arrubia
17. Corrù s'Ittiri
18. Saline di Carloforte
19. Sa Salina
20. Santa Caterina
21. Stano di Porto Botte
22. Stagno di Piscinnì
23. Stagno di Porto Campana
24. Laguna di Nora
25. Stagno di Cagliari
26. Molentargius
27. Codoleddu
28. Colostrai
29. Foce del Flumendosa
30. Sa Praia

Elevation (m asl): 1834, 500, 0

**Figure 1.** Wetland areas of Sardinia (in blue) and location and names of the 30 study sites.

The presence/absence of 323 bird and 844 vascular plant species was recorded for each wetland. Data of both plants and birds were retrieved from all available consulted literature and reports, such as [34]; data repositories, such as wikiplantbase [35] or INaturalist [36], and all available grey literature. These were completely revised by personal data of the authors and, at least, one recent monitoring (2018–2021).

*2.2. Analysis of Species Co-Occurrence*

First, we applied a pair-wise null model analysis to the presence–absence species matrix to determine which associations were significant (i.e., non-random). We used the probabilistic modelling approach developed by [20], which calculates the expected frequency of co-occurrence between each pair of species based on the distribution of one species being independent of the second one. It then compares the expected frequency to the observed frequency and returns the probability that a lower or higher value of co-occurrence could have been obtained by chance. A species co-occurrence analysis was carried out using the function "cooccur" in the cooccur R package [37]. Starting from community data organised in bird and plant species by site matrix, the function returns a list containing pair-wise species co-occurrence results. The package returns a probability of co-occurrence, which can be interpreted as *p* value and classifies species pairs into categories of negative or positive associations. In this case, significant probabilities of co-occurrence were retained upon a threshold of 0.05 [38]. The percentage of significant co-occurrences of each *taxon* were treated as the response variable to be regressed. The average of the same percentages of all the species present in each site was used as the response variable at site level.

*2.3. Species and Site Traits Data*

Following the order of the enumerated hypotheses, we considered the following site- and species-based traits:

H1, environmental filtering hypothesis: in order to test how non-random co-occurrences vary among sites according to their environmental richness and decrease with the presence of threats/impacts, we measured six variables: (1) surface area in hectares; (2) % of open water; (3) distance of the polygon centroid from the coast; (4) number of habitats sensu Habitat Directive 92/43/EEC, retrieved from field observations and literature resources. The level of disturbance was measured by means of (5) protection level, and (6) number of human impacts observed in the field and reported following the MedWet scheme [39]. Because a given site can be protected by more than one designation, e.g., a natural park included in the Natura 2000 network, the level was the result of a sum of different terms of protection, ranging from 0 (no protection) to 2 (maximum protection). For further details, see Supplementary Material (Table S1). Wetland delineations, number of habitats and human impacts are available from the online database of the Italian wetlands (www.italiaiswet.it, accessed on 1 March 2022). The polygons were then processed in GIS environment for measuring the rest of the parameters.

H2, dispersal hypothesis: the dispersal syndrome of each bird species showing non-random co-occurrences with plants was classified as disperser/non-disperser according to the literature concerning the species or its most closely related one. If not directly reported, the information was extrapolated by evaluating their diet and behaviour. The dispersal syndrome of each vascular plant species was retrieved from available databases [40,41].

H3, vegetation structure hypothesis: birds were classified as nesting/non-nesting in Sardinia according to the large amount of available grey and scientific literature (e.g., [42–44]), by only including species regularly nesting on plants in the study sites. Plants were classified as to whether their habit was suitable for bird nesting. The presence of exotic plants was also considered under this hypothesis, because the introduction of alien plants often results in a changed vegetation structure. This information was retrieved from the latest checklists of the Italian alien flora [45]. All traits are reported in Tables S2 and S3.

*2.4. Statistical Analysis*

All statistical procedures were performed in R software (v 3.4.3). Both site- and species-based predictors were analysed by generalised linear models (GLMs) with identity link and Gaussian error distribution to calculate the relative effects on the response of co-occurrence in each site and for each species. In a further step, all analyses were re-run following the application of Cook's distance criterion [46]. This was performed to remove models which

were unduly influenced by a single data point. Two sites ("Codoleddu" and "Stagno di Cagliari") had a Cook's distance greater than one and thus were removed as outliers. Model selection was based on Akaike information criterion corrected (AICc) for small sample sizes. The R package "glmulti" was used [47] to retain, among all possible combinations, only the top-ranked models with AICc differences (ΔAICc) < 2 [48]. The averaged coefficients of the retained models were then calculated using the same R package. *p* values derived from the retained GLMs were coupled with the sum of Akaike weights to estimate the importance of each variable [48]. The variable importance output gives the total weight of each driver across all possible models. The sum of Akaike weights have values ranging from 0 to 1; values close to 1 indicate drivers that occur in large portions of the models and with higher probability that a driver is important [49]. A cutoff of 0.8 was set to differentiate between essential and nonessential predictors [50]. To measure the deviance explained, adjusted D-squared (D$^2$) was also calculated for each top-ranked GLM using modEvA R package [51]. To summarise, the three hypotheses were plotted according to the sum of Akaike weights accounted by each respective predictor retained in the best models.

## 3. Results

### 3.1. Summary of Co-Occurrences among Site-Based and Species-Based Traits

Of the 237,098 interspecific possible pair combinations, 226,193 (95% of the total) were not considered because the species pairs were showing an insignificant probability of co-occurrence (*p* values > 0.05). From the remaining 10,905 pairs, a further 9556 pairs were removed because the expected species co-occurrence was <1 and considered as random. The last 1349 plant–bird pairs including 136 plant and 146 bird species were thus analysed. The highest percentage of non-random co-occurrences was presented for two coastal wetland systems: "Stagno di Sal'e Porcus" (CW Sardinia, 7.8%) and "Stagno di Sa Praia" (SE Sardinia, 7.6%). The lowest percentage of non-random co-occurrences was in the inland temporary pond of "Codoleddu" (SE Sardinia, 2.8%). Among plant species, the lentisc (*Pistacia lentiscus*, 18.9%) and the common reed (*Phragmites australis*, 15.9%) were found to have the highest rates of co-occurrence with birds. The Black-winged Stilt (*Himantopus himantopus*, 19.2%) and the Common Moorhen (*Gallinula chloropus*, 19.1%) were the two birds' species with the highest plant–bird percentage of co-occurrence. See further information concerning each site and species co-occurrence in Tables S1–S3.

### 3.2. Influence of Site- and Species-Based Traits on Plant–Bird Co-Occurrences

When considering the percentage of co-occurrence by the mean of sites' environmental conditions, the best models (ΔAICc < 2) contained two main drivers: area and % of open water, with a sum of Akaike weights across all models of 0.98 and 0.86, respectively (Table 1 and Figure 2). At the bird level, our findings retained two models with ΔAICc < 2 (Table 1). Nesting birds and bird dispersers were the two significant predictors (both with sum of Akaike weights = 0.99). At the plant level, only one predictor trait was in the three retained best models: habits suitable for nesting (sum of Akaike weights = 0.99; Table 1 and Figure 2).

According to these results, each hypothesis was explained by at least one predictor (Figure 2). H1, the environmental filtering hypothesis, was supported by two essential predictors (area and % of open water). H2, the dispersal hypothesis, was confirmed by bird dispersers. In addition, H3, the vegetation structure hypothesis, was supported by two essential predictors (nesting birds and plants with habits suitable for nesting.

**Table 1.** GLM models and information for site-, bird- and plant-based co-occurrence. Best GLMs (all models within ΔAICc ≤ 2) are shown. Predictor traits not present in the model are indicated by NA. See Materials and Methods for explanation of model terms. Adjusted D-squared ($D^2$) was reported in addition to the Akaike information criterion corrected for small sample sizes (AICc). Note that the variable importance, in terms of sum of Akaike weights, was calculated for all possible models. The degrees of freedom (df) are reported for each model. Significance level: *** $p \leq 0.001$, ** $p \leq 0.01$, * $p \leq 0.05$.

| Explanatory: site co-occurrence | | | | | | | | |
|---|---|---|---|---|---|---|---|---|
| **Model terms** | | | | | **Model performance** | | | |
| **Area (ha)** | **N_Habitat** | **D_coast (km)** | **Open Water** | **Protection** | **df** | **Dsq** | **AICc** | **delta** |
| −0.00 ** | 0.07 * | −0.03 * | 0.02 * | −0.99 | 23 | 0.66 | 66.86 | |
| −0.00 ** | | −0.03 * | 0.02 * | | 24 | 0.47 | 67.05 | 0.19 |
| −0.00 ** | 0.06 * | | 0.03 *** | | 24 | 0.45 | 68.12 | 1.26 |
| −0.00 ** | | | 0.00 *** | −0.01 | 24 | 0.45 | 68.56 | 1.51 |
| −0.00 ** | 0.05 | | 0.03 *** | −0.91 | 23 | 0.48 | 68.70 | 1.65 |
| −0.00 ** | | −0.05 *** | | | 25 | 0.41 | 68.73 | 1.87 |
| −0.00 ** | | −0.02 | 0.02 * | | 23 | 0.48 | 68.80 | 1.94 |
| **0.98** | 0.35 | 0.66 | **0.86** | 0.46 | | | | <– Akaike weights |

| Explanatory: bird co-occurrence | | | | | | |
|---|---|---|---|---|---|---|
| **Model terms** | | | **Model performance** | | | |
| **Disperser** | **Nesting** | **df** | **Dsq** | **AICc** | **delta** | |
| 1.70 *** | 2.71 *** | 142 | 0.19 | 739.63 | | |
| **0.99** | **0.99** | | | | | <– Akaike weights |

| Explanatory: plant co-occurrence | | | | | | |
|---|---|---|---|---|---|---|
| **Model terms** | | | **Model performance** | | | |
| **Nest** | **Seeding** | **Exotic** | **df** | **Dsq** | **AICc** | **delta** |
| 2.31 *** | | | 132 | 0.13 | 640.34 | |
| 2.20 *** | 0.47 | | 132 | 0.13 | 641.73 | 1.39 |
| 2.26 *** | | 0.41 | 131 | 0.13 | 642.13 | 1.79 |
| **0.99** | 0.32 | 0.28 | | | | <– Akaike weights |

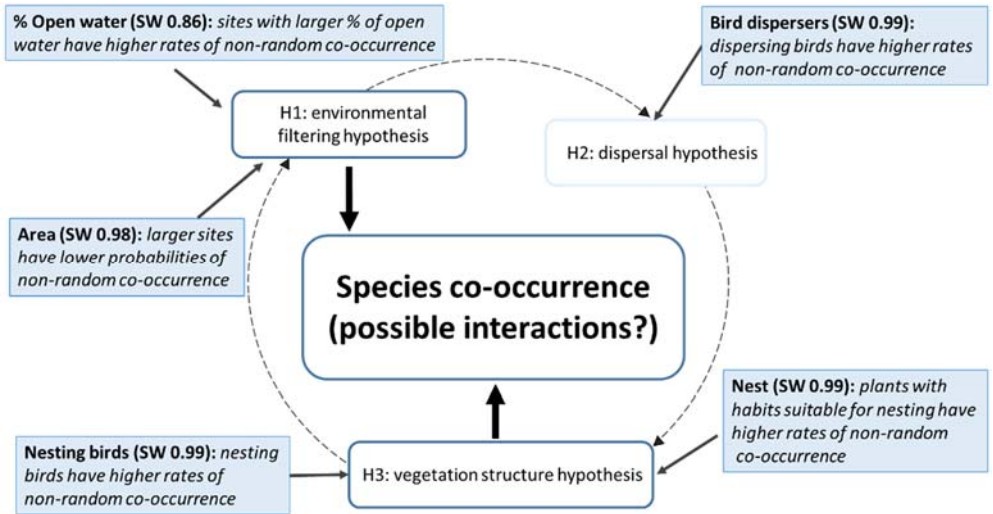

**Figure 2.** Interdependence of the three hypotheses (H1–H3) in determining species co-occurrences at site-, bird- and plant-level. Each hypothesis is supported by predictors with different variable importance (sum of Akaike weights, SW). Accordingly, the three hypotheses were ranked as contributors of the final species co-occurrence rates and possible species interactions.

## 4. Discussion

The example reported by [28] was here confirmed: from an initial relatively large dataset, a large proportion of the co-occurrences were insignificant and thus not valid to efficiently measure the co-occurrence between two species groups. This confirms that, although there is no specific size prescription, many samples are required, much more than what is typically used, for instance, to model single-species distributions. Correlation does not mean causation, and co-occurrence is not evidence of ecological interactions. However, weak and less weak evidences from our results were in line with the hypotheses of biological interactions raised in the literature and corroborated by several experiments.

According to H1, the environmental filtering hypothesis, we found that non-random co-occurrences among sites were influenced by their environmental characteristics. Specifically, co-occurrences and hypothesised interactions decrease with the site extent and increase with the percentage of open water. Although the area plays a prominent role in favouring species diversity [52,53], incongruent patterns in cross-taxon relationships were here found. First, the correlation of species richness between birds and plants is highly variable, and this is likely increasing in disturbed and fragmented habitats [54]. Moreover, habitat fragmentation has been proven to have little effects on network structure, especially for bird species, due to their ability to fly across habitat boundaries [13,55]. Further reasoning can be based on how the distribution patterns of each species group and their functional traits directly vary across space. Previous studies have shown that bird specialisation on specific plant partners increases towards sites with relatively uncommon and extreme environmental conditions, such as forests above 3000 m asl [13] or small islands [50]. Similar specialised plant–bird interactions might be present in small and isolated wetlands where, according to the Italian island inventory [30,56], rare and specialist plant and bird species were more common than in large wetland systems, which are also often characterised by monodominant vegetation communities and relatively low niche variability. If considering that the extent of a site increases with the % of open water, our results are partially in contrast. However, although a few bird species may benefit from naturally vegetated waters, such as the ones dominated by reeds (e.g., [57]), it often negatively impacts on several birds, especially when they are dominated by non-native species, such as *Arundo donax* or *Eichhornia crassipes*, which can lead to loss of diversity and interactions between plants and other species groups [58,59]. Caution might be thus warranted when, such as in our case with *Acacia saligna* (see Table S2), positive co-occurrences are found between some birds and invasive plants. Although some mutualistic interactions between invasive and native—even endangered—species are reciprocally present (e.g., [60,61]), plant–bird interaction networks are generally impacted by the introduction of non-native species [17,62,63].

Little evidence in support of the H2 dispersal hypothesis was found, despite the large body of literature reporting interactions between seed-dispersing birds and dispersed plants (e.g., [7,8]). Only dispersing birds have higher percentages of non-random co-occurrences than non-dispersing birds, whereas this was not found for zoochorous plants. The importance of the avian zoochory for the angiosperms within and beyond a wetland landscape was already illustrated by several authors ([9] and references therein); differently, zoochorous plants might rely on unspecialised dispersers [64]. Moreover, most of the considered dispersal traits were based on assumptions that might be further investigated. In this sense, the need for deeper knowledge was already evidenced [9,65].

Finally, H3, the vegetation structure hypothesis, was also supported by the results. Plant-nesting birds and plants with habits suitable for bird nesting were significant predictors of co-occurrence with the respective other species' group. Our findings suggested that this hypothesis, already largely demonstrated through different approaches (e.g., [5,13,44,66]), might also be confirmed by network reconstructions based on species co-occurrence. Plants are key structural elements of terrestrial ecosystems and thus determine habitat configuration for many animal species, including birds. Even if this interaction might appear rather generalist, that is birds do not need specific plant species for nesting, and plants do not take any specific advantage from nesting birds, it might reveal several

species-specific positive and some non-standard interactions. For instance, seeds can remain attached or embedded within materials gathered for nest building [67], and faeces, fallen nest material and carcass remains left below trees elevate levels of nutrients available to plants in the otherwise poor soils [68].

To summarise, the results obtained from the perspective of conservation management suggest that: (1) interactions between birds and plants, and ecosystem connectivity more in general, are difficult to detect and infer from presence–absence data. However, improving methods in their identification is crucial because their maintenance will ensure the ecosystem self-sustenance. (2) Ensuring a natural vegetation structure is crucial, for birds and for plants conservation. (3) Interactions related to plant dispersion exist, but often in our case were species unspecific and difficult to detect with presence–absence data. (4) Caution is needed in the interpretation of results from presence–absence data because deeper investigations are essential to confirm any real species interaction. However, the high replicability of this simple and low-consumption approach has potential utility for initial screening of the most promising species for further investigation.

## 5. Conclusions

These results suggested that co-occurrence data can provide insightful clues to biotic interactions, although these need to be confirmed by analyses that consider in detail all possible specific situations and conditions. Moreover, co-occurrences might reveal "false-positives" and conceal several interaction types, such as birds promoting the spread of invasive plants and vice versa. The need of a large set of data is again confirmed. Although the initial set of 237,098 interspecific possible pair combinations appears consistent, it was weak considering the low number of significant interactions that emerged from it. However, this work represents a small drop in the ocean of the era of big data, which might be in any case verified and cleaned of spurious and erroneous information. Automated bioinformatic instruments are nevertheless promising.

Ecologists have already largely documented the inherent difficulty in inferring interactions from descriptive data, and many authors are actively engaged in developing methods to do so, while recognising the difficulties. However, network reconstructions based on species co-occurrence are proven as useful to provide insightful clues on phenomena such as co-existence and ecosystem functioning [69]. In closing, we acknowledge that, as with most approaches in science, our results need refinement and improvement. However, the approach has shown sensible predictions that can be widely tested and replicated to increase the knowledge of the hypotheses derived from their application.

**Supplementary Materials:** The following supporting information can be downloaded at: https://www.mdpi.com/article/10.3390/d14040253/s1, Supplementary Material: Explanation of the protection index. Table S1: Co-occurrences and traits related to each wetland site; Table S2: Co-occurrences and traits related to plant species. See Materials and Methods for explanations of each trait. Species names were verified according to The Plant List. Link: https://www.catalogueoflife.org/, accessed on 3 November 2021. Dispersal traits were retrieved according to TRY plant trait database (https://www.try-db.org/TryWeb/Home.php, accessed on 9 February 2022) and LEDA plant traits database (https://uol.de/en/landeco/research/leda, accessed on 10 February 2022); Table S3: Co-occurrences and traits related to each bird species. See Materials and Methods for explanations of each trait.

**Author Contributions:** M.F., A.C.-L. and G.B. conceived the project ideas; M.F. and A.C.-L. designed the methodology; M.F., A.C.-L., C.Z., S.N. and G.B. collected the data; M.F. analysed the data; M.F. and A.C.-L. wrote the manuscript; M.F., A.C.-L., C.Z., S.N. and G.B. validated the data and revised the manuscript. All authors gave final approval for publication. All authors have read and agreed to the published version of the manuscript.

**Funding:** This research was funded by MAVA Foundation.

**Institutional Review Board Statement:** Not applicable.

**Informed Consent Statement:** Not applicable.

**Data Availability Statement:** Details regarding data supporting the reported results can be found in the Supplementary Materials.

**Acknowledgments:** The authors thank the MAVA Foundation for the support to protect wetlands across the Mediterranean area. We also thank Martin Berry for the linguistic advice. We are grateful to the two anonymous reviewers who provided valuable suggestions to improve the manuscript and the editorial board for their work.

**Conflicts of Interest:** The authors declare no conflict of interest.

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
