# Peer review of "Investigating Plant–Bird Co-Occurrence Patterns in Mediterranean Wetlands: Can They Reveal Signals of Ecosystem Connectivity?"

_diversity, doi:10.3390/d14040253_

Round 1
Reviewer 1 Report
Review of the paper "Are plant-bird co-occurrences signals of ecological interactions in Mediterranean wetlands?"
The manuscript is interesting, although the text needs much effort to be reader-friendly. Below, you will find some suggestions to correct the manuscript.
Title needs to be rephrased. The plant-bird co-occurrence is a general single phenomenon. This is a type of ecological interaction, so it cannot signal ecological interactions but rather ecosystem connectivity.
Bird-plant co-occurrence should be used as a general and thus single phenomenon even if it concerns various birds and plants.
Abstract and introduction should be rewritten. In these parts, English is poor, as well as wording is sometimes not appropriate. For example, biotic interactions is not a process, mechanisms cannot be controversial, as well as I do not understand the phrase "data demanding". I do not understand what the significant co-occurrence is - it is rather not random. The phrases about a small drop in the ocean and the era of big data are not acceptable. What do you mean by plant-animal correlations, underappreciated relationship, etc.? Please, spend more time with abstract and introduction.
Hypotheses are interesting. However, you should use rather a term of various ecosystems connectivity resulting from habitat diversity, dispersion including zoochory and searching for nesting habitats.
Analysis of co-occurrence and statistical analysis are reasonable.
Discussion sounds reasonable, although wording is not acceptable in some points, e.g. lessons learned from the results. Summarised results and conclusions are too strong. Please, interpret your results more carefully and be aware of data limitations.
Table S2 and S3 are interesting and could be placed as at least appendix published with the main text.
Author Response
Reviewer 1
The manuscript is interesting, although the text needs much effort to be reader-friendly. Below, you will find some suggestions to correct the manuscript.
Title needs to be rephrased. The plant-bird co-occurrence is a general single phenomenon. This is a type of ecological interaction, so it cannot signal ecological interactions but rather ecosystem connectivity.
RESPONSE: Thank you for the comment. We amended the title accordingly.
Bird-plant co-occurrence should be used as a general and thus single phenomenon even if it concerns various birds and plants.
RESPONSE: Thank you, we have been now considered it.
Abstract and introduction should be rewritten. In these parts, English is poor, as well as wording is sometimes not appropriate. For example, biotic interactions is not a process, mechanisms cannot be controversial, as well as I do not understand the phrase "data demanding". I do not understand what the significant co-occurrence is - it is rather not random. The phrases about a small drop in the ocean and the era of big data are not acceptable. What do you mean by plant-animal correlations, underappreciated relationship, etc.? Please, spend more time with abstract and introduction.
RESPONSE: The entire manuscript, but especially the abstract and the introduction have been revised accordingly. We apologise for the use of unclear phrases/words in the previous version.
Hypotheses are interesting. However, you should use rather a term of various ecosystems connectivity resulting from habitat diversity, dispersion including zoochory and searching for nesting habitats.
RESPONSE: We did not test this hypothesis at the landscape level since the connectivity among ecosystems might be related to several other wetlands that were not included in this analysis.
Analysis of co-occurrence and statistical analysis are reasonable.
RESPONSE: Thank you.
Discussion sounds reasonable, although wording is not acceptable in some points, e.g. lessons learned from the results. Summarised results and conclusions are too strong. Please, interpret your results more carefully and be aware of data limitations.
RESPONSE: We agree with the Reviewer's viewpoint. In our opinion, the caution in the interpretation of our results was already highlighted. However, we have reinforced it, for example, by using the verb "to suggest" instead of "to show".
Table S2 and S3 are interesting and could be placed as at least appendix published with the main text.
RESPONSE: Many thanks. Yes, these tables are included in the supplementary material or appendix and they will be available along with the published paper.
Reviewer 2 Report
This is a very original and well-written paper having a strong rationale and a good statistical design (with strong hypotheses and analyses). Its structure is a bit complex (and the fig. 2 is wellcome!) but the logic is clear. I think that this new version is largely improved and I think that, now, the ms deserves to be published after MINOR REVISIONS. I have only forther minor suggestions.
First. I think that some further reference about birds of Sardinian wetlands (also general papers) should be added. For example, Grussu published many other papers (a part the paper yet cited).
Second, row 50: recently, some effort have been devoted studying the relationships among disturbance, vegetation, and birds (see the papers on hemeroby of Fanelli in Ecological Indicators). Since the authors provided a good arrangement of examples in bird-plants (also reporting relationships with human-induced changes, see row 54), they should add also references in this regard.
Add the role of anonymous reviewers (and Editors).
I have not any further comments (punctuation is ok; Editorial style in references is ok). English language and style are good.
I would like to read a further revised verison of this good ms.
Have a nice work.
Author Response
Reviewer 2
This is a very original and well-written paper having a strong rationale and a good statistical design (with strong hypotheses and analyses). Its structure is a bit complex (and the fig. 2 is wellcome!) but the logic is clear. I think that this new version is largely improved and I think that, now, the ms deserves to be published after MINOR REVISIONS. I have only forther minor suggestions.
RESPONSE: We are grateful for these valuable insights and comments.
First. I think that some further reference about birds of Sardinian wetlands (also general papers) should be added. For example, Grussu published many other papers (a part the paper yet cited).
RESPONSE: We agree that several other researches from Sardinia are present. We added some further references accordingly. Thank you.
Second, row 50: recently, some effort have been devoted studying the relationships among disturbance, vegetation, and birds (see the papers on hemeroby of Fanelli in Ecological Indicators). Since the authors provided a good arrangement of examples in bird-plants (also reporting relationships with human-induced changes, see row 54), they should add also references in this regard.
RESPONSE: We have added the mentioned works. Thank you.
Add the role of anonymous reviewers (and Editors).
RESPONSE: Done, we included it in the Acknowledgments section.
I have not any further comments (punctuation is ok; Editorial style in references is ok). English language and style are good.
RESPONSE: We appreciate your time and inputs, which indeed will improve the quality of this work. Many thanks